



# Dynamical downscaling and data assimilation for a cold-air outbreak in the European Alps during the Year Without Summer 1816

Peter Stucki[1,2,*] and Lucas Pfister[1,2,*], Yuri Brugnara[1,2,**], Renate Varga[1,2,***], Chantal Hari[1,3,4], Stefan Brönnimann[1,2]

[1] Oeschger Centre for Climate Change Research, University of Bern, Bern, 3012, Switzerland
[2] Institute of Geography, University of Bern, Bern, 3012, Switzerland
[3] Physics Institute, University of Bern, Bern, 3012, Switzerland
[4] Wyss Academy for Nature, University of Bern, Bern, 3011, Switzerland
[*] these authors contributed equally to this work
[**] now at Empa, Dübendorf, 8600, Switzerland
[***] now at REWE International AG, Wiener Neudorf, 2355, Austria

*Correspondence to*: Lucas Pfister (lucas.pfister@unibe.ch)

**Abstract.** The "Year Without Summer" of 1816 was characterized by extraordinarily cold and wet periods in Central Europe, and it was associated with severe crop failures, famine, and socio-economic disruptions. From a modern perspective and beyond its tragic consequences, the summer of 1816 represents a rare occasion to analyze the adverse weather (and its impacts) after a major volcanic eruption. However, given the distant past, obtaining the high-resolution data needed for such studies is a challenge. In our approach, we use dynamical downscaling, in combination with 3D-variational data assimilation of early instrumental observations, for assessing a cold-air outbreak in early June 1816. We find that the cold spell is well represented in the coarse-resolution 20th Century Reanalysis product, which is used for initializing the regional Weather Research and Forecasting Model. Our downscaling simulations (including a 19th-century land-use scheme) reproduce and explain meteorological processes well at regional to local scales, such as a foehn wind situation over the Alps with much lower temperatures on its northern side. Simulated weather variables, such as cloud cover or rainy days, are simulated in good agreement with (eye) observations and (independent) measurements, with small differences between the simulations with and without data assimilation. However, validations with partly independent station data show that simulations with assimilated pressure and temperature measurements are closer to the observations, e.g. regarding temperatures during the coldest night, for which snowfall as low as the Swiss Plateau was reported, and a rapid pressure increase thereafter. General improvements from data assimilation are also evident in simple quantitative analyses of temperature and pressure. In turn, data assimilation requires careful selection, preprocessing and bias-adjustment of the underlying observations. Our findings underline the great value of digitizing efforts of early instrumental data and provide novel opportunities to learn from extreme weather and climate events as far back as 200 years or more.



## 1 Introduction

In Central and Western Europe, the extraordinary year of 1816 was referred to by historians as a "Year Without Summer", with particularly cold, wet, and cloudy conditions during the summer months; it is also known as "Eighteen hundred and froze to death" for similar weather and climate in the northeastern US (Auchmann et al., 2012; Briffa et al., 1998; Brönnimann and Krämer, 2016; Crowley et al., 2014; Stommel and Stommel, 1983; Wetter et al., 2011). The shifted precipitation patterns and summer cooling can partly be explained by the enormous and devastating eruption of Mount

Tambora in Indonesia in April 1815 (Fischer et al., 2007; Harington, 1992; Oppenheimer, 2003; Raible et al., 2016; Robock, 2000, 2007; Schurer et al., 2019; Stothers, 1984; Wagner and Zorita, 2005) and, to a lower extent, by random internal variability and low solar variability during the Dalton minimum (Anet et al., 2014). Besides diseases and a socio-economic depression after times of war, the adverse climatic and meteorological conditions led to delayed plant growth, crop failures, poor fruit harvests, rising food prices and famine (Brázdil et al., 2016; Krämer, 2015; Luterbacher and Pfister, 2015; Trigo et

al., 2009). Central Europe and Switzerland were among the most affected regions.

From a modern perspective and beyond its tragic consequences, the Year Without Summer represents one of the few opportunities, or even the only opportunity, to understand the climatic and meteorological situations and developments after a major volcanic eruption, but also to model the potential consequences of such adverse, cold and rainy weather. However, 1816 was more than two hundred years ago, and it is therefore not easy to obtain the necessary information.

The Year Without Summer of 1816 in (Central) Europe has been investigated by historians and historical climatologists, using information from descriptive sources, weather observations and early instrumental measurements (Auchmann et al., 2012; Brázdil et al., 2016; Brugnara et al., 2015; Harington, 1992; Pfister, 1999; Trigo et al., 2009). Most research uses monthly aggregated information to describe the summer of 1816, even if daily resolved observations or measurements are available. For instance, Brugnara et al. (2015; based on Dobrovolný et al., 2010) found that 1816 had the coldest summer in

Central Europe in instrumental records and the second-coldest since 1500 when including documentary evidence. Some studies have analyzed the weather in 1816 also at a daily resolution, mainly on the basis of instrumental observations. An analysis of twice-daily observations of temperature and cloudiness from Geneva, compared to derived weather types, revealed an increased number of cloudy days leading to a larger cooling effect in the afternoon than at sunrise, and an association with more frequent precipitation during the summer of 1816 (Auchmann et al., 2012) than in a contemporary

reference period. Whereas many of these traditional reconstructions of historical extreme events rely on historical weather observations of good quality and within a relatively dense spatio-temporal network (Maugeri et al., 1998; Wagner et al., 2016), modern applications in industry and science increasingly require numerical data from state-of-the-art products that typically provide gridded data.

One of those are atmospheric reanalyses that span the full globe and up to multiple centuries, such as the current Twentieth

Century Reanalysis version 3 (20CR; Slivinski et al., 2019). For the summer of 1816, the first such dynamical atmospheric analysis with a scout version of 20CR was very successful in mapping pressure fields and the associated atmospheric



dynamics over much of the eastern North Atlantic and Europe, despite of a limited network of station barometer observations (Brohan et al., 2016). Most reanalyses provide subdaily, gridded data of a three-dimensional atmosphere, but their spatial resolution is often too coarse to reproduce local processes in the atmosphere and at the surface. Particularly for

modelling the impacts of weather conditions at the surface, data with high spatial resolution is required. For instance, weather reconstructions with horizontal grid sizes at a low kilometer-scale were applied for modelling impacts of (extreme) events on agriculture (Blanco-Ward et al., 2019; Flückiger et al., 2017; Glotter et al., 2014), storms and related economic loss (Pinto et al., 2010; Stucki et al., 2015), or flood events and its impacts on infrastructure (Boé et al., 2007; Mahoney et al, 2022; Rössler and Brönnimann 2018; Stucki et al., 2018), among others.

For Switzerland, gridded daily fields (e.g. of temperature and precipitation) with spatial resolutions as high as 1 km have been created with an analog resampling method (Flückiger et al., 2017; Imfeld et al., 2023; Pfister et al., 2020), and by other statistical approaches including principal component analyses (PCA) of a modern dataset and interpolation of PCA scores from historical station data (Isotta et al., 2019; Stucki et al., 2020). Many of these statistical(-dynamical) reconstructions (or reanalyses) which numerically estimate the state of the atmosphere at a certain point in time involve data assimilation

procedures (Carrassi et al., 2018). On a regional scale, that is with finer temporal and spatial resolution, long-term reanalyses (-derived) products have been produced for several areas by statistical downscaling from global reanalysis products (Caillouet et al., 2016, 2019), and have been further refined by assimilation of local, independent data such as historical surface observations of e.g. temperature or pressure (Devers et al., 2020, 2021).

In this study, we aim to apply so-called dynamical downscaling, combined with assimilation of pressure and temperature

observations, to refine the information on the weather in summer 1816 from the 20CR global, long-term reanalysis products to regional-to-local and (sub-)hourly scale using the Weather Research and Forecast model (WRF; Skamarock et al., 2019). Dynamical downscaling procedures include nesting a limited-area domain from a weather forecast model into the global reanalysis product. This process can then be iterated to refine the global fields of atmospheric variables to local scales (Gómez-Navarro et al., 2018; Michaelis and Lackmann, 2013; Stucki et al., 2015, 2018, 2020). In contrast to statistical

downscaling procedures, physical processes are inherent to the weather forecast model leading to physically consistent simulations of the weather. Physical consistency of the high-resolution simulations is crucial, especially when they are used in a model chain (Maraun et al., 2010, Muerth et al., 2013). Furthermore, statistical downscaling is usually limited to a handful of atmospheric and surface variables, whereas dynamical downscaling simulations provide an encompassing set of variables and preserve their physical coherence (Fowler et al., 2007; Muerth et al., 2013). These properties make dynamical

downscaling attractive despite the high computational costs associated with running a regional circulation model. For nowcasting and forecasting applications in modern periods, weather simulations from downscaling data of global circulation models or reanalyses have been further improved by assimilating additional, local data from independent conventional observation sites, e.g. precipitation observations, and Doppler radar or satellite data (Ban et al., 2017; Fatmasari et al., 2019; Gopalakrishnan and Chandrasekar, 2018; Thiruvengadam et al., 2020; Wang et al., 2013). Whereas such modern remote

sensing techniques were not yet available in 1816, there is a considerable amount of station observations that can be





assimilated, particularly for Central Europe. This opportunity motivates us to explore whether a combination of dynamical downscaling and assimilation of local temperature and air pressure observations can be at all feasible and successful for this region and for such a distant period like the Year Without Summer.

Our simulations focus on an earlier cold spell over Central Europe, which occurred approximately between 5 and 11 June
1816, and for which we have collected information from station measurements of temperature and pressure, but also from weather diaries and records of eye observations regarding sunshine and cloudiness, occurrence of precipitation, wind, and other variables. Being one of the most pronounced cold spells of this summer and with abundant data available, it serves as an excellent example for our analyses. We briefly discuss the representation of our case study period in the 20CR dataset, not least with respect to the feasibility of using 20CR data as atmospheric boundary conditions for the WRF simulations.
Downscaling results are evaluated based on qualitative weather descriptions, as well as quantitative records of surface pressure and temperature. To provide an independent assessment, data from four stations are not assimilated, but retained for validation purposes.

The article is organized as follows: the available station observations, the 20CR reanalysis product, the weather forecast
model, and the data assimilation system are described in Sect. 2. In Sect. 3, we start with a brief description of the weather conditions in the summer of 1816, and more concretely, during the cold spell in June 1816 from documentary evidence and observations, as well as its representation in the 20CR reanalysis product. In the second part, results from dynamical downscaling with the WRF model are shown. This includes (i) simulations without and (ii) with data assimilation of early instrumental station observations. We illustrate the representation of the cold spell in early June 1816 in these simulations
and compare them to independent station observations. A summary and conclusions are given in Sect. 4.

## 2 Data and models

### 2.1 Observations

For the summer of 1816, a large number of surface station observations — at least by the standards of the early instrumental period — is available for Europe (see **Tables 1 and 2**). A large amount of pressure measurements is compiled in the
International Surface Pressure Databank Version 4.7 (ISPD; Compo et al., 2019; Cram et al., 2015; see **Fig. S1** in the Supplement). These observations were assimilated in 20CR, among them are three stations in the area of the European Alps: Geneva, Turin and Hohenpeissenberg. However, the majority of available station measurements of pressure and also temperature from this area – many of those have been digitized only recently – have not been assimilated so far. This opens the door for using these independent observations in the WRF data assimilation procedure. Daily to sub-daily measurements
of temperature and pressure in Switzerland come from the CHIMES project (Brugnara et al., 2020; Brugnara, 2022a), which is based on a digitizing effort in Swiss archives (Pfister et al., 2019). Eight out of the 70 records (at 40 locations) cover the region of interest for the period from 5 to 12 June 1816 **(**see **Table 1**). Additional information from the records for Bern



(observer Studer) were digitized by CHIMES, then quality checked, adjusted and made available by Hari (2021). The station network was completed with further observations of temperature and pressure from recently digitized stations in the Alpine

area (see Brugnara et al., 2015, 2023, **Fig. S2** in the Supplement and **Table 1**). Additionally, there is a good number of European stations outside the greater Alpine region (see **Table 2**), for which pressure and partly also temperature data are available (Brugnara et al., 2015). The station series were quality-checked manually prior to being used for data assimilation and validation. This included checking the metadata (e.g. coordinates, altitude, observation time), converting data to modern units, and tracing gross errors in the data (e.g. temperature too high by an order of magnitude). However, none of the used

observation series are homogenized. This means that whereas homogenization is not required for our series lengths of less than 10 days, each station and each instrument may have erroneous measurements of unknown magnitude. To correct for biases in the measurement series, a simple correction approach was applied (see Sect. 2.4).

For our data assimilation experiment, we only used pressure and temperature data. Further available variables such as wind

velocity and direction, precipitation type and occurrence, fresh snow, and cloud cover were used for validation purposes. As visible in Tables 1 and 2 and **Fig. S2** in the Supplement, all major regions north and south of the Alps contribute data for assimilation in the area of WRF simulations with the highest resolution, with the exception of eastern France. Note again that we used pressure data that had already been assimilated in 20CR. We justify re-using these data for regional data assimilation by the fact that in the reanalysis, they serve to adjust the atmospheric state at low resolution, whereas in our

experiment, they represent a much smaller region. They have less weight next to other stations, while they still provide very valuable information within our small network, especially on local effects that are resolved in the WRF model. Furthermore, individual measurements are often discarded if they show local effects not resolved in the reanalysis. For instance, the (bias-corrected) observation of pressure at Hohenpeissenberg for 6 June 1816 0630 UTC was 885.70 hPa, the lowest but plausible value in our period of interest; it was judged to be too far from the first-guess pressure (- 9.2 hPa; threshold in 20CR is 3.2

times the root of the sum of the squared variances of background and observation) by the algorithm and was thus not assimilated in 20CR (cf. **Fig. S1** in the Supplement).

For an independent validation of pressure and temperature in our model simulations, we use the stations Delémont in the Swiss Jura and Augsburg in southern Germany. In addition, there are parallel time series from independent observers in Bern

and Zurich respectively (Bern Fueter vs. Bern Studer; Zurich Feer vs. Zurich Escher). In principle, these are also independent. Thus, we have two stations located very close to assimilated station records and two stations that are located further away from the nearest assimilated station. On the one hand, a set of four stations is very small for relevant validation, but on the other hand, this allows us to use most of the rare and very valuable information for assimilation.



**Table 1: Weather stations with (assimilated) observational data of pressure and temperature between 5 and 12 June 1816. Independent records used for validation are marked with an asterisk.**

| Station | Lon | Lat | Elevation | Variables | Readings | Digitized period | Source | Remarks |
|---------|-----|-----|-----------|-----------|----------|------------------|--------|---------|
| | ° E | ° N | m a.s.l. | | hour (UTC) | | | |
| Aarau | 8.043 | 47.393 | 384 | T, p | 04, 13, 19 | 1807-01-01 – 1865-12-31 | Brugnara, 2022a | |
| Augsburg* | 10.899 | 48.377 | 494 | T, p | 06, 13, 20 | 1815-01-01 – 1817-12-31 | Stark (1817, 1818a, 1818b) | |
| Avignon | 4.8 | 43.95 | 22 | T, p | 05, 09, 11, 13 | 1816-01-01 – 1816-12-31 | Brugnara et al., 2015 | irregular readings |
| Bern | 7.452 | 46.948 | 534 | T, p | 07, 13, 22 | 1779-12-20 – 1827-07-09 | Brugnara, 2022a; Hari, 2021 | Studer |
| Bologna | 11.353 | 44.497 | 74 | T, p | 11 | 1815-01-01 – 1817-12-31 | Brugnara et al., 2015 | |
| Delemont* | 7.343 | 47.365 | 432 | T, p | 08, 14, 22 | 1801-12-22 – 1832-12-30 | Brugnara, 2022a | |
| Geneva | 6.151 | 46.199 | 396 | T, p | 04, 13 | 1799-01-01 – 1821-12-31 | Auchmann et al., 2012 | |
| Hohenpeissenberg | 11.016 | 47.801 | 995 | T, p | 06, 13, 20 | 1814-01-01 – 1818-12-31 | ISPD 4.7 | |
| Karlsruhe | 8.404 | 49.008 | 121 | T, p | 07, 14, 21 | 1815-01-01 – 1817-12-31 | Brugnara et al., 2015 | irregular readings |
| Marschlins | 9.58 | 46.96 | 562 | T, p | 08, 13, 22 | 1782-01-01 – 1863-11-30 | Brugnara, 2022a | |
| Milan | 9.183 | 45.467 | 132 | T, p | 04, 16 | 1814-01-01 – 1818-12-31 | Brugnara et al., 2015 | |
| Padova | 11.869 | 45.402 | 31 | T, p | 07, 14, 20 | 1815-09-23 – 1817-12-31 | Brugnara et al., 2015 | |
| Rovereto | 11.05 | 45.9 | 200 | T, p | 07, 15 | 1782-02-13 – 1839-08-27 | Brugnara et al., 2023 | |
| St. Gall | 9.379 | 47.425 | 676 | T, p | 03, 19 | 1813-01-01 – 1853-06-07 | Brugnara, 2022a | |
| Schaffhausen | 8.639 | 47.696 | 400 | T, p | 06, 14, 22 | 1794-01-01 – 1845-10-28 | Brugnara, 2022a | |
| Turin | 7.68 | 45.1 | 281 | T, p | 04, 12, 19 | 1787-12-01 – 1865-06-30 | Brugnara et al., 2015 | p only at noon |
| Vevey | 6.844 | 46.46 | 378 | T, p | 07, 22 | 1805-12-01 – 1840-08-15 | Brugnara, 2022a | irregular readings |
| Zurich | 8.544 | 47.372 | 427 | T, p | 10 (?) | 1816-06-01 – 1816-12-31 | Brugnara, 2022a | Escher |
| Bern* | 7.452 | 46.948 | 534 | P | 07, 22 | 1805-03-15 – 1833-11-21 | Brugnara, 2022a | Fueter |
| Zürich* | 8.544 | 47.372 | 418 | T, p | 07, (13), 21 | 1807-01-01 – 1827-12-31 | Brugnara, 2022a | Feer, irregular readings |




**Table 2: Additional weather stations within outermost WRF domain with (assimilated) observational data of pressure and temperature between 5 and 12 June 1816.**

| Station | Lon | Lat | Elevation | Variables | Readings | Digitized period |
|---|---|---|---|---|---|---|
| | ° E | ° N | m a.s.l. | | hour (UTC) | |
| Althorp | -1.000 | 52.280 | 105 | T, p | 08, 20 | 1816-01-01 – 1817-11-30 |
| Armagh | -6.648 | 54.353 | 64 | T, p | 08, 12, 14, 20 | 1815-01-01 – 1818-02-28 |
| Barcelona | 2.173 | 41.383 | 20 | T, p | 07, 14, 22 | 1814-01-01 – 1818-12-31 |
| Barnton | -3.292 | 55.962 | 50 | T, p | 12 | 1815-01-01 – 1817-12-14 |
| Boston | -0.028 | 52.977 | 10 | T, p | 13 | 1816-01-01 – 1817-12-31 |
| Coimbra | -8.424 | 40.210 | 95 | T, p | 08, 09, 10, 11, 12, 13, 14, 15, 16, 17, 18, 19, 20, 21 | 1815-01-01 – 1817-05-31 |
| Exeter | -3.529 | 50.723 | 47 | T, p | 08, 09, 14, 15, 22 | 1814-01-01 – 1817-12-31 |
| Gdansk | 18.653 | 54.349 | 14 | P | 05, 13, 21 | 1815-01-01 – 1817-12-31 |
| Goteborg | 11.966 | 57.705 | 15 | T, p | 05, 06, 13, 21 | 1815-01-01 – 1817-12-31 |
| Haarlem | 4.650 | 52.383 | 2 | T, p | 08, 13, 22 | 1814-01-01 – 1818-12-31 |
| Krakow | 19.956 | 50.064 | 212 | T, p | 05, 13, 20 | 1816-01-01 – 1816-12-31 |
| London | -0.117 | 51.517 | 24 | T, p | 07, 08, 14, 15, 16 | 1815-01-01 – 1817-12-31 |
| Lviv | 24.032 | 49.842 | 295 | T, p | 02, 03, 04, 05, 06, 12, 20 | 1815-03-01 – 1817-09-15 |
| Paris | 2.337 | 48.836 | 65 | T, p | 09, 12, 15, 21 | 1816-01-01 – 1817-12-31 |
| Prague | 14.417 | 50.083 | 202 | T, p | 13 | 1815-01-01 – 1817-12-31 |
| Rochefort | -0.963 | 45.933 | 25 | T, p | 08, 15 | 1815-01-01 – 1818-12-31 |
| Stockholm | 18.050 | 59.350 | 44 | T, p | 01, 02, 03, 04, 05, 06, 07, 08, 12, 13, 19, 20 | 1814-01-01 – 1818-12-31 |
| Uppsala | 17.641 | 59.861 | 15 | T, p | 01, 02, 03, 04, 05, 06, 07, 08, 13, 14, 15, 17, 20 | 1814-01-01 – 1818-12-31 |
| Valencia | -0.376 | 39.474 | 25 | T, p | 07, 13, 18 | 1815-07-08 – 1818-12-31 |
| Vaxjo | 14.803 | 56.877 | 170 | T, p | 05, 13, 21 | 1815-01-01 – 1817-12-31 |
| Vienna | 16.350 | 48.233 | 198 | T, p | 07, 14, 21 | 1815-01-01 – 1817-12-30 |
| Zitenice | 14.162 | 50.553 | 223 | P | 04, 05, 06, 13, 20 | 1815-01-01 – 1818-12-31 |
| Zwanenburg | 4.733 | 52.383 | 5 | T, p | 07, 08, 13, 22 | 1814-01-01 – 1818-12-31 |




## 2.2 Reanalysis product

The NOAA-CIRES-DOE Twentieth Century Reanalysis version 3 (20CR; Slivinski et al., 2019) is used for synoptic analyses and as initial and boundary conditions for the downscaling experiments. 20CR is a 4-dimensional global dataset that
provides 8-times daily fields of atmospheric variables on a ~75-km horizontal grid from 1836 to 2015; a publicly available experimental extension with all 80 ensemble members goes back to 1806. The latest version of 20CR represents a substantial refinement of former versions which provided 4-times daily fields with ~100 km horizontal resolution and temporal coverage from the mid-19th century onwards (Compo et al., 2011).  Here, we mainly use the mean of the 80 ensemble members.

## 2.3 Regional circulation model

The non-hydrostatic Advanced-Research Weather Research and Forecast Model version 4.1.2 (WRF-ARW; WRF hereafter; Skamarock et al., 2019) is used for dynamical downscaling from 20CR. The three nested, limited-area domains have cell sizes of 27, 9, and 3 km, and the grid sizes are 127x109, 211x184, and 256x220 cells. The innermost domain is relatively large to avoid complex mountainous terrain at the boundaries, where possible. There are 60 eta levels in the vertical with a
top level of 50 hPa. The model calibration builds upon previous WRF downscaling applications over the same region (Dierer et. al., 2014, Gomez-Navarro et al., 2015, Stucki et al., 2015, 2016, 2018, 2020) The Thompson microphysics scheme (Thompson et al., 2008) is used for bulk microphysical parameterization, the Yonsei University (YSU) scheme (Hong et al., 2006) for the planetary boundary layer. The Kain–Fritsch scheme was used for cumulus parameterization in the larger domains (Kain, 2004), and turned off in the innermost domain. Spectral nudging (corresponding to a wavelength of about
1000 km) is applied to temperature, wind, and geopotential fields above the planetary boundary layer in the 27-km domain for consistency with large-scale forcing (von Storch et al., 2000). The use of spectral nudging has been analyzed and recommended in multiple studies, particularly for terrain with marked orography (Feser et al., 2011; Liu et al., 2012; Ma et al., 2016; Spero et al, 2014, 2018). In order to not restrain the simulation too much towards the large scale forcing, the nudging coefficients were set to 0.0001 (Stauffer and Seaman, 1990). The WRF model is initialized on 4 June 1816 00 UTC,
allowing for approximately 24 hours of model spin-up before the cold spell starts. The simulation datasets are stored in hourly resolution.

Most of the above settings mostly correspond to common standards or even operational specifications, with exceptions. One of our more elaborate tests addressed the effects of using the standard modern-time land use scheme from the United States
Geological Survey (USGS) vs. a reclassification of Anthromes v2 (Anthropogenic Biomes version 2; Ellis et al., 2010), which provides land use categories for the year 1800. One interesting feature appeared in that nightly temperature drops were more moderate in places where the local land-use changed to more urban conditions in modern times (**Fig. S3** in the supplement); the nights are simulated up to 5 °C warmer in the built-up areas compared to cropland and woodland. The





median shift during the night is 1.8 °C for Bern (3.6 °C for Geneva, 2.6 for Aarau), while the afternoon hours are around

0.25 °C warmer. Apart from this, only small changes in variables such as albedo, latent heat flux or total column cloud

fraction occurred in all three domains. Based on this, the historical land use scheme was used as the standard configuration.

## 2.4 Data Assimilation System

In addition to dynamical downscaling *without* data assimilation described above, WRF simulations were combined with a

3D-Var data assimilation system (WRFDA; Skamarock et al., 2019). Simulations *without* (*with*) data assimilation are called

NODA (DA) hereafter. The WRF setup for the DA simulations is the same as for NODA simulations, including spectral

nudging in the outermost domain. With this combination of spectral nudging and subsequent regional data assimilation, we

follow a number of previous studies that have already successfully adopted this technique for dynamical downscaling. For

their regional 15-km Arctic System Reanalysis, Bromwich et al. (2018) implemented nudging on temperature, geopotential

height and wind at wavelengths > 1000 km before using WRFDA for conventional observations, among others. Lin et al.,

(2021) combined spectral nudging with 3D-Var assimilation of radar data for precipitation forecasts and Yao et al. (2021)

used atmospheric and snow data assimilation for springtime temperature simulations. For further details on the WRFDA

system, see Barker et al. (2004, 2012), Huang et al. (2009) and Skamarock et al. (2019; their chapter 11 and references

therein).


Our basic idea for the assimilation was to mimic modern surface synoptic observations (SYNOP) of air pressure and

temperature. The conversion into SYNOP implied assigning specific observation times for each series. Whereas a number of

records include time indications on (sub-) hourly scale, indications of the time of the day like 'noon' or 'sunset' were

transformed using the R package suncalc (Thieurmel and Elmarhraoui, 2022). To mitigate heterogeneity of observations and

ensure compatibility with the 20CR input dataset, we implemented a simple bias-correction using a second harmonics fit

(temperature) and a running mean difference (pressure) using data from the nearest grid cell and time step from 20CR,

vertically interpolated to station elevation. This bias correction was done for the whole year 1816. While for temperature,

each time of the day (e.g. 00 UTC, 03 UTC, etc.) was corrected separately, pressure was corrected using measurements from

a moving window of 15 timesteps. Potential timing errors were compensated by the bias correction procedure.

In order to tune the WRF data assimilation system, a variety of analyses have been performed (not shown), including

assimilation of raw data, assimilation of temperature only in the innermost domain, and a range of values for the observation

errors and perturbation scaling. However, sensitivity studies in a technical sense were not possible given the exploratory,

'proof-of-concept' nature of this study. In the applied data assimilation system, measurement errors were set to 1 K for

temperature and 1 hPa for pressure (given bias-corrected observations) after some experience with measurement rejections

by the WRFDA system. Observations are rejected if the innovation exceeded the observation error by a factor of 10 or for an

altitude difference of more than 200 m. The background error covariance matrix was calculated using differences from 12h

and 24h forecasts for the same timesteps (see Parrish and Derber, 1992) over the period May – July 1816. The analysis was



calculated every 3 hours with an assimilation window of $\pm$ 1.5 hours around the analysis timestep. Note that observations for assimilation were not available for all time steps; in these cases, empty station files had to be fed to WRFDA. As stated

above, a limited set of tests regarding error estimates, data assimilation parameters and bias-correction has been performed (not shown). In all, we consider the described WRFDA configuration as a subjectively optimal trade-off between enabling freedom of the model and restraining heavy dependence on observations, which would be problematic due to possible errors in the data.

## 3 Results

### 3.1 The meteorological situation in June 1816

In a first step, we describe the summer of 1816 from available studies and observations, and in particular, the cold-air outbreak over Central Europe and the Alps between 5 and 11 June 1816. Long-term weather records by observers in Geneva, St. Gall, and Bern, Switzerland (**Fig. 1**) show that June 1816 was among the five rainiest and clearly the least sunny month between 1799 and 1821 (Auchmann et al., 2012; Brugnara et al., 2022b; Hari, 2021). The only day with reports of "plenty of

sunshine" was 3 June 1816. Temperatures during the month of June were far below seasonal average; measured afternoon temperatures in Geneva were below 10 °C during the first ten days of the month (Auchmann et al., 2012, see their Fig. 2), leaving them among the lowest between May and October. Furthermore, it was extremely cold in the south-eastern Alps: in Rovereto (Brugnara et al., 2023) 7 and 8 June 1816 were by far the two coldest summer days of the 1800-1839 period, i.e. colder than the next coldest day by 3 °C and 2 °C, respectively. According to historical information compiled by Pfister

(1999), among others, frequent northwesterly flow in Switzerland led to long rainy episodes and persistent cloudiness, especially from 4 to 21 June. In fact, subdaily reconstructions of temperature and (interpolated) pressure by Brugnara et al. (2015; **Fig. S4** in the Supplement) depict the Alpine region to the southeast of a surface high over the British Isles and a surface low over southern Scandinavia during this episode (see also Brohan et al., 2016) and show a temperature and pressure gradient along the Alps. Accordingly, documentary information report intermittent snowfall throughout June, even

at low elevations (Pfister, 1999). Four days of snow below 1500 m a.s.l. were observed between 6 and 10 June 1816, and snow as low as 500 m a.s.l. (at Weggis, Lake Lucerne) was seen on 6 June. Digitized records from Bern (observer Studer; Hari, 2021), Aarau (Zschokke) and St. Gall (Meyer) for 1816 support this information: May, June and July each had >20 days with precipitation, 5 to 11 June 1816 were all rainy days, and snow and rain was observed on 6 June 1816 in St. Gall. Further reports from the CHIMES raw material (Brugnara et al., 2020) include snowfall in Bern on 8 June, and hail on 11

June (observer Fueter), and snowfall in Delémont on 6 June 1816 (unknown observer). Note that during the same period, also the northeastern US and Canada were hit by snowstorms and a cold wave (Chenoweth, 2009).



**Figure 1: Processed and original information from selected weather diaries in Switzerland. a) Classes of cloud cover (grey symbols for cloud fraction, from overcast (index 1) to mixed (0.5) to sunny (0) conditions) and amounts of precipitation (blue filled circles; larger circles mean more precipitation) derived from twice-daily observation records in Geneva (Auchmann et al., 2012) for all months (day of month in x-axis) of June between 1799 and 1821. The right margin summarizes instances of clear-sky conditions (where larger grey circles indicate brighter conditions) and of precipitation for each month. b) As in a), but with four classes of cloud cover (grey symbols) and three categories of precipitation (blue symbols) as derived from twice-daily observation records in St. Gall (Meyer; CHIMES) for all months 1816. c) As in a), but with four classes of cloud cover (grey symbols) and observations of precipitation (blue filled circles) from twice-daily observation records in Bern (Studer; Hari, 2021) for June 1816.**



Next, we aim to assess how well 20CR can reproduce and plausibilize the synoptic weather situation that we have outlined from the traditional reconstructions. In the first place, **Fig. S1** in the Supplement shows that the spatial coverage of the North-Atlantic region with instances of pressure information from ISPD is good, i.e. compared to other regions of the world

and this point in time. Note that from this pressure information, the assimilation into 20CR generates a set of 80 ensemble members as deviations from the ensemble mean. **Figure S5** in the Supplement delineates the 1005-hPa isobars over the North-Atlantic region for all 80 members and the ensemble mean in 20CR for 6 June 1816 18:00h. Overall, the SLP contour line of the ensemble mean is well located within the range of ensemble members, and the pressure minimum over the Alps is also well within the bulk of the members. Similarly, the ensemble mean of a 2-meter temperature time series, extracted for a

grid cell over Switzerland (7.7° E, 47° N), runs with the bulk of the members. Although slightly smoothed with regards to the ensemble variability, it clearly reflects the cold episode for this region.

In fact, 20CR indicates that the cold spell over Central Europe started with a deep cored low over Scandinavia on June 3 (not shown). While the low slowly retrograded in a south-westerly direction, cold air masses were trapped over the subpolar eastern North Atlantic and trajectories show a south(-east)ward transport of cold air between this cyclonic system and a ridge

to the west of the British Isles (not shown). On June 6, a marked pressure gradient formed over the Alps, and the coldest air masses (around or below freezing point) reached the north side after sunset, whereas temperatures remained significantly higher south of the Alps due to *foehn* effects **Fig. 2a**). In the following days, the steering low made one more cyclonic pivot over Scandinavia to re-establish itself west of Denmark on June 8 - 9. This led to a shift of the advection to (south-)westerly over central Europe, then to a rather calm situation and then back to northerly on June 10. From here, the position of the

Scandinavian low shortly renewed the cold air advection towards the Alps before weakening and giving way to an extending Azores High on June 11 (not shown).

These analyses with the current 20CR version 3 can be seen as a continuation of the pioneering, pre-20CR reanalyses by Brohan et al. (2016). They were already able to indicate higher pressure west of the British Isles and a low-pressure system over southern Scandinavia. However, their assimilation was based on a very limited network of station barometer

observations. Thus, they indicated limited skill for the eastern North Atlantic. Furthermore, the synoptic situation resembles the one described by Brugnara et al. (2015) for July 1816, which points to repeated patterns of a zonal pressure dipole over northern Europe causing cold-air advection over Central Europe and other European regions.

Hence, the current 20CR version 3 represents a big step forward, although our synoptic analyses still reveal coarse spatial patterns, and the ensemble mean may not necessarily concur better with the real atmospheric state than some individual

ensemble members. It clearly captures the dynamics of a cold-air outbreak in two episodes between 5 and 11 June 1816 with a peak in the night from June 6 to 7. This underpins the plausibility of the analysis and hence the quality of 20CR for this region and the early 19[th] century. In all, this also means that the 20CR ensemble mean is an adequate basis for the next step, the dynamical downscaling.



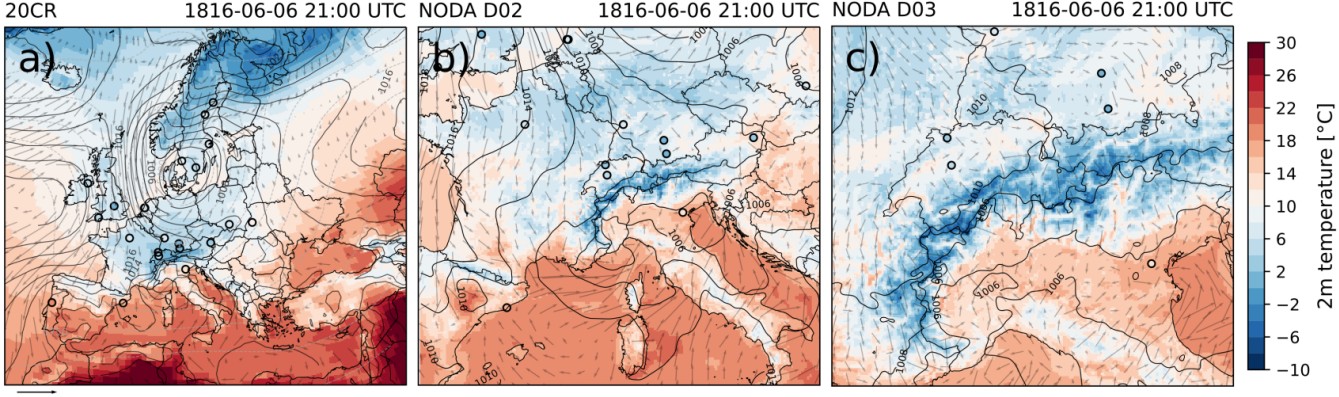

**Figure 2: Analysis of (a) - (c) temperature at 2 meters above ground (shade: degrees Celsius), 10-meter wind (grey vectors; reference vector approx. 10 ms⁻¹), mean sea level pressure (black contours; hPa) and geopotential height at 500 hPa (light grey dashed contours, meters) over Europe and the Alps on 6 June 1816 21:00 as calculated from 20CR in (a) and the WRF domains 02 (b) and 03 (c). Station observations of temperature are indicated as filled circles.**

**3.2 The cold period 5 – 11 June 1816 in the WRF NODA and DA simulations**

In this section, we explore the potential of WRF to produce more detailed weather maps that plausibly reflect the cold-air outbreak. For this, we zoom in from global to regional and local scales, following the three nested WRF model domains from the outermost to the innermost. With the refinement of the global information to regional scales, areas with below-zero temperatures at night become more distinct along the Alpine mountain ranges, central and northeastern France, western Germany, the Pyrenees and Cantabrian Mountains, and even southern England (not shown). For instance, analyses of the

time step 6 June 1816 21:00 in domain 01 (not shown) and 02 (**Fig. 2b**) from the NODA simulation reflect and refine the predominant northerly flow of cold air masses from the North Sea to the Alps, and a split of the flow over northeastern France. Finally, domain 03 (**Fig. 2c**) incorporates the full Alpine bow and adjacent regions to the north and south. At this scale, local weather conditions are simulated at a horizontal grid size of 3 km. With a first swath of cold air reaching the north side of the Central Alps on 5 June 1816 (not shown), areas of very low temperature are simulated on elevated Alpine

terrain (e.g. below -20 °C in Valais), but also on the Jura mountains and along the Alpine foothills during the following nights (e.g. from 6 to 7 June 1816). In contrast, many valleys and areas south of the Alps were even warmer than 10 °C. The same pattern is also evident from the temperature observations (**Fig. 2a-c**), although some biases with respect to 20CR and the NODA simulations are visible. This refined pattern of a north *foehn* situation becomes apparent from the SLP field, which delineates a north-south gradient across the Alps. Associated were northerly winds (>10 m/s) across the Alpine rim

(note even some counterflow) and a marked meridional temperature difference with comparatively warmer conditions south of the Alps. Accordingly, the distinct cloud layers at low and mid-levels on the northern side of the Alps were dissipated and bright skies appeared on the southern side (**Fig. 3a**). Over the following days, the northerly flow changed to westerly and southwesterly from 8 to 9 June 1816 (not shown) to the north and west of the Alps, while the secondary low took shape over



Northern Italy (**Fig. 3b and d**). This induced substantially warmer air on both sides of the Alps, especially during day-time.

On 11 June, the weather situation changed again (**Fig. 3c**): The Adriatic surface low weakened while the Azores High extended, and an associated low established over the North Sea (not shown). On the meso-scale, this brought a shift from westerly to calm, then increasingly north-easterly winds and air flow to the north side of the Alps.

**Figure 3: Analysis of (a) - (c) cloud cover (cloud fraction from 0 to 1) at low atmospheric levels (L), mid-levels (M), and high levels (H) as well as (total column) precipitable water (red dashed contour lines at 25 kg/m2), sea level pressure (smoothed dashed lines; hPa), and wind (grey vectors, every 8th in each direction is shown) (d) as in Figure 2c. Simulations are with WRF NODA domain 03 for instances in time indicated on top of each panel.**

Most of these spatial patterns and weather features come out similarly from the NODA and DA simulations (not shown). In contrast, the cold spell and a daily cycle of very low temperatures for the season are reproduced somewhat differently in the



two simulations. **Figure 4 (a - c)** shows temperature maps for the coldest point in time near sunrise on June 7. The temperature maps show a considerable difference between the NODA and DA simulation. The DA simulation has lower temperatures by up to 4 °C for most of the domain, and up to 10 °C in the inner-Alpine region.

**Figure 4: Top: 2m temperature maps for 7 June 1816 at 04:00 UTC. Shown are results for a) the NODA run, b) the DA run, and c) the DA increment. Light grey arrows indicate the wind field. Station data for the corresponding assimilation window are indicated as colored circles. The straight line indicates the extent of the cross section. Bottom: cross section (model orography is grey) of air temperature (shade; °C) from the Po plain (8.25° E, 45° N) to the Vosgues mountains (6.85° E, 48.5° N) and up to 4000 m. a.s.l. for 7 June 1816 04:00 UTC. Shown are cross sections for d) NODA and e) DA simulations. Freezing level and the 3°C-Isotherm are emphasized as bold black lines.**

The significantly lower temperatures in the DA simulation are also reflected in a cross section of the temperature field perpendicular to the Alpine bow, i.e. from the Po plain near Turin, Italy to the Vosgues mountains near Colmar, France, for the same instance in time (**Fig. 4d and e**). For this early morning, freezing levels drop from around 2000 m a.s.l. just near the Alpine rim to around 1200 m a.s.l. further north in NODA. Analogously, the 3 °C isotherm (approximately equivalent to the transient snowline on the ground) lies at around 1200 m a.s.l. at the northern alpine flank and drops to values to below 1000 m a.s.l. further to the north. Generally, both simulations are in line with the reports of recurrent snowfall below 1500 m





a.s.l., arguably including snow and rain in or near St. Gall (around 800 m a.s.l.). However, only the DA simulation indicates

freezing below 1000 m. a.s.l. and a 3 °C isotherm reaching the lowest elevations north of the Alps. Thus, only the DA simulation does closely fit with reports of snow as low as 500 m a.s.l. or even lower, for instance at Bern (Swiss Plateau), Delémont (Jura), or Lake Lucerne (Alps). From this, we assume that the vertical temperature profile might be a little too warm in the WRF NODA simulation. We also infer that DA results in a more realistic representation of temperature, and that in general, our WRF simulations are able to reflect the effects of the large-scale cold-air outbreak on a local level. This leads

us to a more detailed comparison of the simulations with observations for specific locations in order to gain a more general view over the differences between the NODA and DA simulations.

### 3.3 Verification of NODA / DA simulations with systematic observations and measurements

We assimilated as much of the very sparse station data as possible to obtain the most plausible simulation in terms of spatial and temporal dimensions. For this reason, we retained only four independent stations for point-by-point verification of

temperature and pressure. Two of them are quasi-parallel measurements (around 1 km apart) in Zurich and Bern and two stations are at least 50 km away from the next assimilated station. Due to these limitations, our verification is performed as follows. In a first part, we consider variables that are independent of the simulation: Precipitation, cloud cover, radiation, and wind can be compared with eye observations of, e.g., cloud cover, occurrence of rain, or sunshine and wind direction. In a second part, we include temperature and air pressure with a particular emphasis on the independent station records. Finally,

we summarize the results with a quantification of the potential improvements from DA. First, we show the comparisons of observations with simulations for four locations in Switzerland from southwest to northeast, which are Geneva, Bern, Zurich and St. Gall (**Fig. 5**). Comparisons with the WRF NODA / DA simulations are done using the grid point closest to the respective station location. Simulated pressure and temperature values were thus converted to station elevation for the quantitative assessment.

Cloud cover observations from the Swiss stations agree about fairly bright skies for 4 June only. From there, intermittent overcast days are reported but they are not coincident across the four stations. Note in this context, that there are inconsistencies between cloud cover and reports of rain in the observations for some stations. The station of Zurich has two measurement series of relative humidity. While the one by observer Feer appear to have too low values, possibly because measured indoor, the readings at noon by Escher are among the highest of the summer (average of 81% between 4-12 June

with maximum of 88% on the 6th) and support the observations of dark skies. The simulations produce more consistent cloud cover: 4 June was mostly bright, followed by thick clouds from around 6 to 12 June, but with clearly lighter cloud cover into 8 June. This is mostly consistent with the simulated values of tropospheric relative humidity and the reduced shortwave radiation; they indicate that the darkest days were on 6, 9 and 10 (extending into 11) June. Note also that the finding of increased cloudiness over Geneva in the afternoons by Auchmann et al. (2012) is partly reflected in the simulated

cloud cover.







**Figure 5: Meteograms of station observations and measurements (red colors) for a) Geneva b) Bern c) Zurich and d) St. Gall with WRF NODA (ligher colors, grey, blue, orange) and DA (darker colors) simulations output for the nearest grid point, for the period between 4 and 12 June 1816 (x-axis). Top panels show observed cloud cover (red squares from 'bright' with no fill to 'mixed' with cross and 'covered' with fill) vs. simulated low, mid- and upper level cloud fraction (larger bars indicate more cloudiness). Second row panels show simulations of downward short wave flux at ground surface (orange lines; W/m2) and relative humidity at 700 hPa (dashed blue line; permil). Red crosses (dark for observer Escher, light for Feer) indicate the relative humidity measurements on the ground for Zurich. Third row panels show thrice-daily observations of precipitation (red dots; red vertical bars for Geneva measurements in mm; darker red dots for Zurich stand for observer Escher, lighter for Feer; red cross for 'rain and snow' added for St. Gall on 6 June 1816) vs. simulated precipitation (blue vertical bars, mm). Bottom panels show observations of wind direction (red vectors at unit length; north is up) vs. simulated wind direction (grey wind vectors; north is up) and velocity (black line and vector length; m/s at 10 m above ground).**



The only station with precipitation measurements is Geneva. Qualitatively, these observations agree very well with the simulated patterns of cloudiness, humidity and precipitation and support the notion of the above-mentioned cloudiest days.
The other stations show intermittent rainfall from around 6 June onwards, with snow and rain on that day in St. Gall, but with no break on 8 June, as in the simulations. This leaves us with the understanding that the simulations reproduce the observed weather evolution with brighter days at the beginning, and dark and rainy days from June 6 (with a break on June 8). However, the correspondence is not evident in all details, and some of the observation entries may be inaccurate. For instance, observations of cloud cover were noted twice or three times a day, but at varying times depending on the observer,
and their assignment to a certain hour and the chosen categories by us are not concurrent.

The comparisons of observed and simulated wind further illustrate the qualities and limitations of the observations. Most of the information appears only reliable for periods with stronger winds and is thus hardly exploitable for comparisons. For Bern (and similarly well for Schaffhausen; not shown) however, the wind information supports the simulated evolution of the meso-scale circulation; with a predominant northerly flow until June 9, followed by stronger southwesterly winds for two
days, then a drop in wind speeds and easterly (*bise*) wind directions towards the end of the period. Hence, we find that these particular records are a good example of astonishingly accurate observations, given that measuring highly variable wind parameters in an adequate quality is a difficult task, even with modern infrastructure and thoughtful site selection.

Differences between NODA and DA are mostly small to negligible, with the exception of slightly higher humidity and more
precipitation in the DA simulation. In fact, the accumulated precipitation over the investigated period and across the Swiss Plateau is higher in DA (by up to around 20 mm; **Fig. S6** in the Supplement), while it is lower over the parts of the Alps. This might be attributed to the assimilation of (in tendency lower) temperature and pressure values over the Swiss Plateau and its possible effects on the formation of convective precipitation in the simulation.. Due to the lack of quantitative station data, a thorough assessment of simulated precipitation values is not possible.


In contrast to wind, cloud cover and precipitation, records of surface temperature and pressure were obtained from instrumental measurements at all locations. In addition to the comparisons for Bern and St. Gall (**Fig. 5b and d**), our assessments for these variables also feature the independent station series from Delémont and Augsburg, as well as the two independent parallel records from Bern (observer Fueter) and Zurich (observer Feer; **Fig. 6**). The difference of temperature
readings for the parallel records is between 1 and 2 °C. Note that the bias correction applied to the data observed by Feer is larger than the one for the Escher data. Note also that in the original observations, there is a marked bias compared to simulations for some stations, especially where the elevation of the model terrain and observation site differ considerably (i.e. by more than 50 to 70 m). The reason for this bias is not entirely known and may lie in both simulation or observations. Evidence from the bias-corrected series (with 20CR; see Sect. 2.4), which do not exhibit a large bias, indicate that the latter
is more probable. Warm biases in the order of 1–2 °C were in fact not uncommon in early instrumental temperature measurements during summertime. The surface pressure records from Studer and Fueter in Bern (**Fig. 6c**) agree very well



with each other. The measurements by Fueter are higher by as much as 2.5 hPa, mostly because his values were not reduced to 0 °C. This causes an overestimation of about 2 hPa, which is well within the range of the general uncertainty of SLP at that time of about 5 hPa which can largely be attributed to uncertainties in station elevation (see Brugnara et al., 2015; their

Fig. 8). In fact, Studer even read a second barometer to verify the calibration; the difference between his two barometers in June 1816 is about 0.5 hPa. For the two series from Zurich (**Fig. 6d**), deviations of pressure readings appear larger than for Bern, but with a deviation still within common measurement errors.

Our assessment of air temperature shows that the general agreement between observed and simulated near-surface (2m) temperature is very good for the shown independent stations (**Fig. 6**, note that there's no parallel temperature series available

for Bern). The daily cycle of the simulation mostly touches the measured values, and no general bias is apparent, although there are differences of up to 5 °C at times. For some instances, there are biases between raw and bias-corrected observations, with distinctive patterns depending on the measurement time (e.g. evening measurement at Delemont in **Fig. 6a**). This points to an error in assumed observation times, which was leveled out with the bias correction. Comparing NODA and DA simulations, DA clearly shows lower night-time temperatures, especially for the coldest nights of June 7 and 8. For

June 11, the 2-m temperature from the DA simulation is generally lower compared to NODA, and DA agrees better with the measured temperatures at all stations. Compared to the climatological means for the period 1981 to 2010 (provided by the Swiss Federal Office for Meteorology and Climatology MeteoSwiss), the daily maxima in the historical episode are only near or slightly above the daily minima for the modern period. These results are similar for the other stations (not shown). We conclude from the comparison to the historical measurements as well as the modern climatology that temperature seems

well simulated in general, but some minima and maxima were probably more pronounced than seen in the simulation. Also, the pressure series from all stations agree very well with the simulated values. The pressure information supports the idea of brighter skies at the beginning and end of the period and clearly unsettled weather on June 6 and 10. However, there are substantial differences between observations and the NODA simulation at the two pressure minima on June 6 and 10. The readings differ from each other in time (up to 6 hours), and they are all between 2.5 and 8 hPa lower than in the NODA

simulation. In addition, the subsequent pressure increase is markedly stronger in the observations. The same deviations can also be observed for the other available stations. In the DA simulation, surface pressure still does not reach the observed minima on June 6 and 10, but they are more pronounced for these days (differences between 0 and 5 hPa), and there is a stronger pressure increase after the two observed minima. In short, both simulations reproduce the general evolution of surface pressure, although the variability of pressure over time is arguably too small in the simulations and the temporal

evolution too smooth. Data assimilation led to an improvement in this respect, and the values coincide better with observations than NODA.





**Figure 6: Comparison of independent station records from a) Delemont, b) Augsburg, c) Bern (observer Fueter), and d) Zurich (observer Feer) with WRF simulations for the period between 5 and 11 June 1816 (x-axis). Shown are 2m temperature (left) and surface pressure (right) from the WRF NODA and DA experiments (lines) taken from the nearest grid point to the observation site, as well as raw and bias-corrected observations. Mean bias and root mean squared error (RMSE) between observations (bias-corrected) and the corresponding simulated values is indicated in each graph both for NODA and DA simulations. For reasons of completeness, the assimilated (and thus dependent) station data from Bern (e-f) and Zurich (g-h) are also indicated (grey = raw, black = bias-corrected).**






**Figure 6** shows also that mean bias and RMSE are substantially reduced by DA for the four independent stations and for both temperature and pressure, with very few exceptions. To get more spatial context, we calculated these measures for all available station records, shown in **Fig. 7**. For a majority of 14 out of 18 station records, the median bias of temperature is again substantially lower for the DA simulations compared to NODA. It drops by approx. 0.5 °C for all stations and by 1 °C for the independent stations. Similarly, RMSE values become substantially lower. On average, they drop by 0.2 °C for all stations, and by 0.4 °C for the independent stations (**Fig. 7A-b**). Surface pressure biases (**Fig. 7c**) indicate an increase of the median bias by approx. 0.5 hPa for all stations, and an enhancement (reduction) by around 0.3 hPa for the independent stations. The RMSE (**Fig. 7d**) decreases by approx. 0.4 hPa with DA for all stations, and by approx. 0.6 hPa for the independent stations. Overall, both non-independent and independent stations show the same patterns of improvements from DA. This reinforces our confidence that the spatial context of the local improvements is correct and that more independent stations would arguably show similar improvements. Furthermore, the differences of the quasi-parallel measurements in Bern and Zurich show approximate margins of error, which seem highly acceptable in a qualitative respect.

Note that the RMSE and the associated deviations from simulations are particularly high for temperature and pressure values from Marschlins and Turin, as well as temperature readings from Hohenpeissenberg and pressure readings from Aarau and Rovereto, arguably pointing to quality issues of the mentioned station series. In the case of Rovereto, the large pressure differences were found to arise from an uncertainty in observation times (not shown). The clearest outlier is the station of Marschlins. The data from this station were difficult to use in several respects. On the one hand, the data seem to be questionable (e.g. there are measurements of almost 30 °C; possibly on a sunny wall?), on the other hand, conversions into today's units of measurement could be erroneous, and thirdly, the station is located on a valley slope where the elevations for the station itself as well as in the WRF topography are very uncertain (errors of >= 200 m are possible).

From these analyses we can take that although state-of-the-art forecast verification cannot be done in our historical context, the comparisons reveal very good agreement between observation and simulations. For pressure, this might be expected as pressure is assimilated in 20CR. Although deviations between observations and simulations are within the observation error of early instrumental readings, simulated pressure variability is too smooth compared to the observations, especially in the NODA simulation. One reason for this might be the use of the ensemble mean from 20CR as atmospheric boundary conditions; using individual ensemble members might improve the simulated surface pressure. Generally, also temperature observations agree well with the simulations. Our quantitative assessments show that data assimilation clearly improves the results regarding both temperature and pressure. Strongly erroneous records are likely rejected by the assimilation algorithms, such that the simulation accuracy is not decreased.





**Figure 7: Quantitative evaluation of simulated pressure and temperature at all observation sites. Independent (parallel) records are marked with an asterisk. Shown are bias (a) and root mean squared error (b) for temperature, as well as bias (c) and RMSE (d) for surface pressure compared against homogenized station series. Annotations on the right hand side of the respective graphs show median biases, as well as the RMSE over all records and over independent records, respectively.**

## 4 Summary and conclusions

The Year Without Summer 1816 was characterized by exceptionally cold spells in Central Europe. First, our analyses describe the meteorological situation of a cold spell over the European Alps in early June 1816 based on traditional reconstructions and in the 20[th] Century Reanalysis. Then, we provide weather simulations on an hourly temporal and 3-km



(horizontal) spatial scale from two experiments: dynamical downscaling (i) without data assimilation (NODA) and (ii) with
3D-Vat assimilation (DA) of bias-corrected pressure and temperature observations from stations in Switzerland and (Central)
Europe. Lastly, simulations are qualitatively and quantitatively compared to available early instrumental measurements and
eye observations.

The cold-air outbreak over central Europe and the associated large-scale northerly air flow between a marked depression
over Scandinavia and a ridge of high pressure west of the British Isles is well captured in the 20CR ensemble mean. The
ensemble mean stands for a middle scenario within the bulk of the 80 realisations from the ensemble members and was
found adequate to use as atmospheric boundary conditions for our downscaling simulations. Among others, this is in line
with previous tests for the same region and the distant past (e.g. Stucki et al., 2018, 2020).

The quality of 20CR opened the field for experiments with dynamical downscaling, i.e. repeated nesting of the regional
weather model WRF into the global 20CR data. Experiments with a land use scheme representing the early 19[th] century
derived from the Anthromes project (Ellis et al., 2010) found differences in night-time temperature for locations that have
been urbanized since, while other variables only showed negligible differences compared to a modern scheme. We thus used
this old land use scheme for our analyses. Our downscaling simulations reproduce regional- to local-scale meteorological
processes such as the *foehn* wind situation across the Alps with much lower temperatures on its northern side.

In general, the downscaled cloud cover, short wave radiation and relative humidity agree well with eye observations of
cloudiness or sunshine conditions. This is less true, of course, for timing and absolute values. For instance, precipitation may
be simulated too conservatively. The simulated evolution of advection is well reflected in some of the available wind
observations. Indeed, some observers were set to deliver astonishingly accurate and meticulous meteorological information.
Our validation analyses with independent (i.e. not temperature and pressure, which were assimilated in DA) weather
variables showed that differences between NODA and DA are mostly negligible or small (e.g. cloud cover, shortwave
radiation, humidity, wind, precipitation). For temperature and pressure (assimilated variables) however, DA simulations are
clearly closer to the observations. For instance, lower pressure minima and a sharper rise after frontal activity are simulated
with DA, whereas pressure variability is too small in the NODA simulations. Colder night-time temperatures, lower freezing
levels and reported snowfall as low as around 500 m a.s.l. are only reflected in DA simulations. The general improvements
with DA are also found in simple quantitative analyses of stations with independent and dependent temperature and pressure
series. A careful selection and bias-correction of the assimilated station records is nevertheless crucial, as their quality
largely affects the DA results.

In all, the analyses show that numerical weather simulations for this region and the early 19th century provide realistic
atmospheric properties and dynamics, at a local, kilometer-scale resolution. Despite a relatively sparse observational network




and a rather short simulation period that does not allow a thorough validation of the general impacts of data assimilation on downscaling reconstructions, we conclude from our assessments that dynamical downscaling results are successfully improved by our data assimilation experiment. The continual historical observations and descriptions, available through digitizing efforts like the CHIMES project, are a prerequisite and extremely valuable for numerical studies of extreme weather and climate events of the past, and for many more scientific purposes and practical applications. In this sense, the aspect of mutual exploitation becomes ever more important: better numerical methods allow the inclusion of more observations and more variables (such as information on wind, cloud cover, rain/non-rain, as shown here), and this, again,

will lead to better (i.e. regional, long-term, high-resolution) reanalyses. With an envisaged extension to climatological timescales, our approach provides novel opportunities for the scientific community to learn from extreme weather and climate events as far back as 200 years; the prospects of soon entering the 18[th] century with such 4-dimensional studies are very good.

**Acknowledgements**

Support for the Twentieth Century Reanalysis Project version 3 dataset is provided by the U.S. Department of Energy, Office of Science Biological and Environmental Research (BER), by the National Oceanic and Atmospheric Administration Climate Program Office, and by the NOAA Physical Sciences Laboratory. We are particularly thankful for the extensive testing of dynamical downscaling (including non-standard land-use schemes) that was done by Andrey Martynov. Our thanks also go to Marcelo Zamuriano, who helped to advance the WRF downscaling. We would not be at this point without

their technical skills and pionneering work. The authors would further like to thank Santos J. González-Rojí for his help setting up the WRF data assimilation system.

**Funding**

Funding for Lucas Pfister, Peter Stucki and Yuri Brugnara was available through the  "Swiss Early Instrumental Meteorological Data" (CHIMES) and the "Long Swiss Meteorological Series" projects, the latter was funded itself the

Global Climate Observing System (GCOS) Switzerland. Lucas Pfister and Peter Stucki were as well funded by the Swiss National Foundation project "Daily Weather Reconstructions to Study Decadal Climate Swings". Yuri Brugnara was also funded by the Copernicus Climate Change Services (C3S) 311c Lot 1 and 2.

**Code and Data availability**

Pressure measurements from the International Surface Pressure Databank Version 4.7 (ISPD; Compo et al., 2019) can be
downloaded from https://doi.org/10.5065/9EYR-TY90. Station measurements related to the CHIMES project are available



from PANGAEA at https://doi.org/10.1594/PANGAEA.948258 and https://doi.org/10.1594/PANGAEA.961277. The 20[th] Century Reanalysis version 3 (Slivinski et al., 2019) ensemble members back to 1836 can be downloaded from NERSC at https://portal.nersc.goy/project/20C_Reanalysis/. The experimental extension of 20CR back to 1806 is obtainable from NOAA upon request. Wrapper scripts and other tools used for running WRF and WRFDA are available at NCAR's WRFDA
users' page https://www2.mmm.ucar.edu/wrf/users/wrfda/download/tools.html.

Setup files for the WRF downscaling experiments are stored on the University of Bern Open Repository and Information System BORIS  (Stucki, 2023; https://doi.org/10.48350/189671)

**Author contribution**

SB, PS, LP planned the campaign; LP performed the WRF simulations; CH, YB, RV, SB, PS provided and processed the
observational data; CH, LP, PS, YB, RV analyzed the data; PS and LP did the visualisations and wrote the manuscript; YB, RV, CH, SB reviewed the manuscript.

**Competing interests**

The authors declare that they have no conflict of interest.

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
