# Peer review of "Dynamical downscaling and data assimilation for a cold-air outbreak in the European Alps during the Year Without Summer 1816"

_EGUsphere, 2023_

## Referee Comment (RC1)

In this study, the authors test the ability of the WRF model to simulate a cold spell over the European Alps during the Year Without Summer (1816). For this purpose, the authors employ two different configurations of the model: a simulation including 3DVAR data assimilation and another one without. Results show that even if the simulation including data assimilation consumes more computational resources and needs a more careful set-up (the available stations must be carefully selected first), it improves the results compared to the simple WRF simulation. Both simulations can simulate the observed general weather conditions, but only the one including data assimilation is closer to observations in terms of temperature and pressure. Thus, the authors highlight the improvements obtained due to the data assimilation only, and the novel opportunities provided by the digitalization of early records to study previous weather events.

The manuscript follows a logical structure, and it fits into the scope of Climate of the Past. However, some major comments need to be addressed by the authors before the manuscript is ready for publication.

**Major comments:**

**Introduction.**
I think that a sentence about the possibility of learning from past extreme events should be added to the paragraph about the objectives of the study. It was briefly mentioned in the abstract, but I think that it should be added reaffirmed here.

**Section 2.1 Observations:**
This section is difficult to follow, and it should be straightforward for the reader. Thus, I suggest clarifying some points from it:
(1)  Are the three stations assimilated in 20CR included in WRFDA again? In Table 1, Geneva, Turin and Hohenpeissenberg are listed, but it is unclear if those are the same as those from 20CR.
(2)  In line 131 is stated that "Eight out of the 70 records (at 40 locations) cover the region of interest", but this information is not included in the tables. I think Tables 1 and 2 should provide specific details about the assimilated observations in WRFDA (total number of records, dates, etc). In the current state, the tables provide general information about the records (digitized period, source, etc.), but not the specific details that could facilitate understanding the DA assimilation in the model.
(3)  Some information about Table 1 is missing: what are the implications of "irregular readings" in the Remarks column? Why some hours of the Readings column are between brackets (e.g., Zurich and Zurich*)?

**Section 2.3 Regional circulation model:**
Some comments about this section:
(1)  I think that the authors should include a figure in which the set-up of the three nested domains is included. I thought that the set-up was that from Figure 2, but then I realized that Figure 2a is a plot from 20CR reanalysis, so I was wondering if

that is still the original parent domain of the WRF simulations. I think that it would be easier if the authors could include a new figure with the three-domain set-up (even in the supplementary).

(2) The spin-up of the simulation is approximately 24 hours. Have the authors checked if that is enough to let the model reach the equilibrium (particularly for the land/soil)?

*Section 2.4 Data Assimilation System*:
Some critical information about the DA is missing in this section:

(1) Were the observations assimilated in all three domains, or only in the bigger domain (D1)?

(2) What method was followed to create the background error covariance matrix?

*Section 3.1 The meteorological situation in June 1816:*
Lines 287-296: This paragraph explains the atmospheric conditions that led to the cold-air outbreak over the Alps in June 1816. However, all the plots from 20CR data that could facilitate the understanding of the event are missing in the manuscript. I think the authors should include a summary of the atmospheric dynamics that triggered the event using the 20CR data, including different panels for different time steps and linking each of them to different lines of the paragraph. This would allow the readers to have in mind an idea of the development of the cold outbreak, and what should they expect from WRF.

*Section 3.2 The cold period 5-11 June 1816 in the WRF NODA and DA simulations:*
Line 320: This is linked to a precious comment for section 2.3 (1). Why domain 1 of WRF is not shown in Figure 2? Wouldn't it be better to include it along the plot for the 20CR input data, so that the reader can see already the improvements in spatial resolution made by WRF (from ~75km to 27 km)?

*Section 3.3 Verification of NODA simulations with systematic observations:*

(1) Lines 378-379: I think that a link to section 2.4 (where the correction methods are explained) should be included here.

(2) Figure 5: I think that a lot of information is included in the same figure, and due to its current resolution, not everything is visible even if you zoom in on the different panels. Thus, I would suggest the authors split it into two figures (Two variables for each station in one figure and the other two variables in another for example to match the structure of the text). Additionally, some labels are not visible (e.g., only one value is visible in the Y-axis of wind and precipitation) or omitted (e.g., labels for relative humidity). Also, it is difficult to differentiate the SW values (orange lines) and the clouds (grey lines) for both WRF simulations.

(3) Figure 6: There is a mismatch between the labels in the plots and the labels in the captions. At the end of the caption, labels e, f, g and h are mentioned, but in the figure, only a-d are marked. Additionally, I think that some corrected values for the assimilated station for Zurich (in black) are not plotted in the Figure (from 06-05 to 06-07).

**Minor comments:**

- References should be listed in chronological order throughout the introduction and the rest of the manuscript (e.g., lines 37-38, 40-41, etc)
- Lines 84-86: the sentence is long and tricky to understand. I would suggest reducing it or splitting it into two different sentences.
- Figure S1: I would add a white background for the legend included in the figure, as it is difficult to read the way it is now. Also, would it not be better to use red and blue dots directly instead of red dots and blue circles over the dots? In the end, all dots are assimilated at some point, right?
- Line 126-127: I think the sentence should be separated into two sentences.
- Line 186: A full stop after the references is missing.
- Line 373: "independent of the simulation". Could it be that it should be independent of the assimilation?
- Line 423: "in the simulation.." Remove one of the full stops.
- Line 480: (Fig. 7A-b). A shouldn't be capitalised.
- Line 488: "and Turin". However, in Fig.7, it is called Torino (also in Fig. S2). Please, agree on a way of calling the stations and keep it throughout the entire paper.
- Line 515: 3D-VAR assimilation.
- Line 529: Be more accurate and specific instead of "old land use scheme"
- Line 534: Be more specific about the comparison of modelled precipitation against observations.

---

## Author Comment (AC1)

**Reply to the reviewers comments (RC1)**

General Remarks for Stucki and Pfister et al. In this study, the authors test the ability of the WRF model to simulate a cold spell over the European Alps during the Year Without Summer (1816). For this purpose, the authors employ two different configurations of the model: a simulation including 3DVAR data assimilation and another one without. Results show that even if the simulation including data assimilation consumes more computational resources and needs a more careful set-up (the available stations must be carefully selected first), it improves the results compared to the simple WRF simulation. Both simulations can simulate the observed general weather conditions, but only the one including data assimilation is closer to observations in terms of temperature and pressure. Thus, the authors highlight the improvements obtained due to the data assimilation only, and the novel opportunities provided by the digitalization of early records to study previous weather events.

The manuscript follows a logical structure, and it fits into the scope of Climate of the Past. However, some major comments need to be addressed by the authors before the manuscript is ready for publication.

We'd like to thank the reviewer for this positive feedback and for all the helpful comments and suggestions to improve the manuscript.

**Major comments:**

*Introduction.*

I think that a sentence about the possibility of learning from past extreme events should be added to the paragraph about the objectives of the study. It was briefly mentioned in the abstract, but I think that it should be added reaffirmed here.

Thank you for this suggestion. We will add such a sentence to the introduction.

*Section 2.1 Observations:*

This section is difficult to follow, and it should be straightforward for the reader. Thus, I suggest clarifying some points from it:

(1) Are the three stations assimilated in 20CR included in WRFDA again? In Table 1, Geneva, Turin and Hohenpeissenberg are listed, but it is unclear if those are the same as those from 20CR.

(2) In line 131 is stated that "Eight out of the 70 records (at 40 locations) cover the region of interest", but this information is not included in the tables. I think Tables 1 and 2 should provide specific details about the assimilated observations in WRFDA (total number of records, dates, etc). In the current state, the tables provide general information about the records (digitized period, source, etc.), but not the specific details that could facilitate understanding the DA assimilation in the model.

(3) Some information about Table 1 is missing: what are the implications of "irregular readings" in the Remarks column? Why some hours of the Readings column are between brackets (e.g., Zurich and Zurich*)?

Thank you for these valuable remarks. We will restructure the section. First, we will describe the threee sources of observational data (ISPD, CHIMES, other digitizing efforts). Then, we will describe the quality control and bias adjustment for regional assimilation. Finally, we will highlight the stations used for validation.

In particular,  (1) the three stations are indeed the same used for assimilation in 20CR as stated in the paragraph starting at line 144. In the revised manuscript, we will make this fact clearer in the text and indicate it also in the remarks in Table 1. (2) the sentence refers to the eight records from the total of 70 digitized series within the CHIMES data rescue project that cover the region (and period) of interest. We will clarify this

in the text and further clarify the data source in Table 1. (3) We will introduce the explanations in the table header.

*Section 2.3 Regional circulation model:*

Some comments about this section:

(1) I think that the authors should include a figure in which the set-up of the three nested domains is included. I thought that the set-up was that from Figure 2, but then I realized that Figure 2a is a plot from 20CR reanalysis, so I was wondering if that is still the original parent domain of the WRF simulations. I think that it would be easier if the authors could include a new figure with the three-domain set-up (even in the supplementary).

Thank you for this suggestion. We will add a plot showing the set-up of the three nested domains, as well as the location of the stations in Tables 1 and 2 in the supplement. To implement the suggestions for improvement regarding figure 2 in this review and in the second reviewer's comments, we will indicate the borders of all three domains in figure 2a and add a panel showing results from the outermost WRF domain. The figure caption will be adjusted accordingly.

(2) The spin-up of the simulation is approximately 24 hours. Have the authors checked if that is enough to let the model reach the equilibrium (particularly for the land/soil)?

Thank you for this important point. Model spin-up time can be a crucial point for accurate downscaling simulations. Whereas spin-up times of around 24 hours are common for shorter downscaling simulations spanning a period of days to weeks (see e.g. Michaelis and Lackmann, 2013), we wanted to verify its effect on the model results following your comment. For this purpose, we created a WRF simulation (without data assimilation) with a long spin-up time of 31 days (initialized on May 4[th] 1816 00:00 UTC) otherwise applying the same WRF setup as described in the manuscript. A comparison of the two WRF simulations with 24h and 31 day spin-up, respectively, yielded the following results:

- Comparisons with station observations show larger temperature differences (instantaneous differences reach maximum values of approx. 4° C) the 5[th] of June 1816. The 24h spin-up simulation leads to an underestimation of measured temperatures compared to the 31d spin-up simulation. This issue is related to a larger Alpine snow cover in the initial conditions (reanalysis) of the 24h spin-up simulation which takes around 48 hours to melt. Temperature differences mostly dissipate until the 6[th] of June 1816, when the most pronounced episode of the simulated cold-air outbreak started. Smaller temperature differences within an expectable range for two realizations of such a simulation (approx. 0.5 °C) are found throughout the rest of the period. Individual stations show sub-daily differences between the simulated temperatures due to cooling during precipitation events. Simulated surface pressure values of both simulations agree very well (max. differences of 0.5 hPa) arguably as a result of spectral nudging. A quantitative evaluation for both temperature and pressure (results shown in the figure below) did not show conclusive result, i.e. there is no clearly better or worse simulation compared with station observations.

[Figure]

- Simulated soil moisture shows minor differences (a few percent) mostly related to slight differences in location and intensity of precipitation events rather than systematic deviations arising from an un-equilibrated soil moisture due to different spin-up times.

- Daily precipitation amounts and patterns of both simulations are similar. The simulation with 31 days of spin-up revealed stronger precipitation on June 6 and 11 whereas differences are negligible for the Alpine area on the other simulated days.

From these analyses, we can conclude that generally – given that the resources are available – longer spin-up times may improve downscaling simulations, or not. For the particular case study of the cold-air outbreak in 90 early June 1816 we see limited benefit from such a longer spin-up time. Whereas differences are restricted mostly to the first two days of the 24h spin-up simulations, both experiments agree well for the remaining simulated period and the quantitative evaluation does not yield conclusive results. Temperatures and their night-time minima which are the main focus in this paper are well captured in both simulations. We therefore decided to remain with the original 24h spin-up simulations for the analyses presented in the manuscript. A 95 sentence that different spin-up times have been assessed will be added in the revised manuscript.

_Section 2.4 Data Assimilation System_:

Some critical information about the DA is missing in this section:

(1) Were the observations assimilated in all three domains, or only in the bigger domain (D1)?

Observations were assimilated in all three domains in order to ensure consistency between the nested domains as much as possible. We will clarify this in the revised manuscript.

(2) What method was followed to create the background error covariance matrix?

The background error covariance matrix has been calculated from differences between 12h and 24h predictions over three months as indicated in the manuscript in line 236f. This procedure has emerged as a standard for the WRF data assimilation system. We will clarify this procedure in the revised manuscript.

*Section 3.1 The meteorological situation in June 1816:*

Lines 287-296: This paragraph explains the atmospheric conditions that led to the cold-air outbreak over the Alps in June 1816. However, all the plots from 20CR data that could facilitate the understanding of the event are missing in the manuscript. I think the authors should include a summary of the atmospheric dynamics that triggered the event using the 20CR data, including different panels for different time steps and linking each of them to different lines of the paragraph. This would allow the readers to have in mind an idea of the development of the cold outbreak, and what should they expect from WRF.

In principle, we agree with the comment, and we had in fact produced a number of figures to illustrate the evolution over time in 20CR. However, we came to the conclusion that the focus of the article should be more on the downscaling and the regional data assimilation rather than on a case study with 20CR, and that we should focus on evidence that the 20CR ensemble mean is adequate for downscaling. For this reason, we would prefer not to include a new figure in the manuscript. However, we will include such a figure in the Supplement: It will show a number of relevant atmospheric variables at three instances in time that are mentioned on page 12, L286ff.

*Section 3.2 The cold period 5-11 June 1816 in the WRF NODA and DA simulations:*

Line 320: This is linked to a precious comment for section 2.3 (1). Why domain 1 of WRF is not shown in Figure 2? Wouldn't it be better to include it along the plot for the 20CR input data, so that the reader can see already the improvements in spatial resolution made by WRF (from ~75km to 27 km)?

Thank you for this comment. We will indicate the borders of all three domains in figure 2a (showing the reanalysis) and add a panel showing results from domain 01 in the revised manuscript.

*Section 3.3 Verification of NODA simulations with systematic observations:*

(1) Lines 378-379: I think that a link to section 2.4 (where the correction methods are explained) should be included here.

Thank you for pointing this out. We will add a corresponding reference in the revised manuscript.

(2) Figure 5: I think that a lot of information is included in the same figure, and due to its current resolution, not everything is visible even if you zoom in on the different panels. Thus, I would suggest the authors split it into two figures (Two variables for each station in one figure and the other two variables in another for example to match the structure of the text). Additionally, some labels are not visible (e.g., only one value is visible in the Y-axis of wind and precipitation) or omitted (e.g., labels for relative humidity). Also, it is difficult to differentiate the SW values (orange lines) and the clouds (grey lines) for both WRF simulations.

We think that the current resolution issues come from the fact that the submitted vector graphic was converted to a raster image for the manuscript discussion version. We are confident that this will be resolved in the final typesetting of the manuscript.

Regarding the content, a similar suggestion was made by Referee 2. Our intention was to show as many variables for comparisons as possible to provide a good picture of the nature and quality of the data. However, we realize that it would be better to reduce the information. For this, we will just keep the Bern location with the full information in one panel, and we will have a second panel with just the most illustrative variables, and without the NODA information on it, see the planned figure caption below. To compensate, we will place the current Figure with the full information in the Appendix. Missing tick labels will be added in the revised figures.

"Figure 5: Meteograms of station observations and measurements (red colors) for a) Bern (station id be01) with WRF NODA (ligher colors, grey, blue, orange) and DA (darker colors) simulations output for the nearest grid point, for the period between 4 and 12 June 1816 (x-axis). The top panel in shows observed cloud cover (red squares from 'bright' with no fill to 'mixed' with cross and 'covered' with fill) vs. simulated low, mid- and upper level cloud fraction (larger bars indicate more cloudiness). The second row panel shows simulations of downward short wave flux at ground surface (orange lines; W/m2) and relative humidity at 700 hPa (dashed blue line; permil). The third row panel shows thrice-daily observations of precipitation vs. simulated precipitation (blue vertical bars, mm). The bottom panel shows observations of wind direction (red vectors at unit length; north is up) vs. simulated wind direction (grey wind vectors; north is up) and velocity (black line and vector length; m/s at 10 m above ground). Panels b) are as in a), but for selected values for Zurich (station id zh00), Geneva (ge00) and St. Gall (sg01). Red vertical crosses (dark for observer Escher, light for Feer) indicate the relative humidity measurements on the ground for Zurich. Red vertical bars show precipitation measurements in mm for Geneva; darker red dots for Zurich stand for observer Escher, lighter for Feer. The red tilted cross means 'rain and snow' for St. Gall. Grey plus signs mean confirmed obserations, minus signs mean no record of precipitation found."

(3) Figure 6: There is a mismatch between the labels in the plots and the labels in the captions. At the end of the caption, labels e, f, g and h are mentioned, but in the figure, only a-d are marked. Additionally, I think that some corrected values for the assimilated station for Zurich (in black) are not plotted in the Figure (from 06-

05 to 06-07).

Thank you for the close look. We will correct this mismatch originating from a former version of the figure in the revised manuscript. For the series of Zurich (observer Escher, indicated as grey/black dots), the available sub-daily series only starts on June 1$^{st}$ 1816 even though this series is known to cover a longer period from published monthly means. Bias-corrected pressure values prior to the 8$^{th}$ of June 1816 are not available due to the bias-correction applying a moving time-window of 15 time steps. The first timestep of this series allowing for the bias-correction is thus the 8$^{th}$ of June. We will mention this issue in the revised manuscript and adjust the indication of the digitized period in Table 1 accordingly.

**Minor comments:**

-    References should be listed in chronological order throughout the introduction and the rest of the manuscript (e.g., lines 37-38, 40-41, etc)

Unfortunately, we are unable to change this. We use the reference formatting tool for Climate of the Past in the Mendeley software, which produces alphabetical orders; see also the author guidelines at https://www.climate-of-the-past.net/submission.html#manuscriptcomposition

-    Lines 84-86: the sentence is long and tricky to understand. I would suggest reducing it or splitting it into two different sentences.

We will reformulate the sentence in the revised manuscript for better readability.

-    Figure S1: I would add a white background for the legend included in the figure, as it is difficult to read the way it is now. Also, would it not be better to use red and blue dots directly instead of red dots and blue circles over the dots? In the end, all dots are assimilated at some point, right?

Thank you for having a look at the Supplement as well. We revised the plot according to your first suggestion, with red and blue dots – this reduces the information a little, but it is much better readable. We refrain from using one color only because the assimilation of the station of Hohenpeissenberg, for instance, is discussed in the article.

- Line 126-127: I think the sentence should be separated into two sentences.

At these lines, we actually see two sentences, so it's hard to see what should be changed. Note also that the whole paragraph will be rewritten due to your major comment on Sect. 2.1.

- Line 186: A full stop after the references is missing.

Thank you for the hint, we will add it.

- Line 373: "independent of the simulation". Could it be that it should be independent of the assimilation?

Thank you for the hint – yes, assimilation makes more sense here.

- Line 423: "in the simulation.." Remove one of the full stops.

We will do so, thank you.

- Line 480: (Fig. 7A-b). A shouldn't be capitalised.

We will change this, thank you.

- Line 488: "and Turin". However, in Fig.7, it is called Torino (also in Fig. S2). Please, agree on a way of calling the stations and keep it throughout the entire paper.

Thank you for pointing this out. We will unify the station names throughout the paper and the supplement.
The journal guidelines suggest using the English names where available.

- Line 515: 3D-VAR assimilation.

We will correct this.

- Line 529: Be more accurate and specific instead of "old land use scheme"

We will change this to: "the 19$^{th}$-century land use scheme".

- Line 534: Be more specific about the comparison of modelled precipitation against observations.

We agree that precipitation deserves more attention. We will add a couple of sentences.

---

## Author Comment (AC2)

**Reply to the reviewers comments (RC2)**

The paper focuses on the cold-air outbreak in early June 1816, during the Year Without Summer, characterized by extraordinarily cold and wet periods in Central Europe. Using the Weather Research and Forecasting (WRF) Model and the 20th Century Reanalysis (20CR) product, the authors perform dynamical downscaling combined with data assimilation of early instrumental observations. The findings highlight the capability of the WRF model to reproduce regional to local meteorological processes and improve the accuracy of simulations when early pressure and temperature measurements are assimilated.

The Year Without Summer, caused by the eruption of Mount Tambora in 1815, has been extensively studied in historical climatology. Previous research often relied on descriptive sources and early instrumental measurements aggregated on a monthly basis. This paper builds on these foundations by providing high-resolution, sub-daily weather simulations, adding significant detail to the understanding of climatic impacts during this period.

The work provides a novel approach to analyzing historical weather events using a combination of dynamical downscaling and data assimilation. This method allows for a more detailed and accurate reconstruction of past weather events than was previously possible, offering new insights into the meteorological conditions of the Year Without Summer. The findings underscore the importance of digitizing early instrumental data and demonstrate the potential of modern numerical models in historical climatology.

The manuscript is well-structured and thorough, presenting a detailed methodology and comprehensive results. The use of both qualitative and quantitative validation against independent historical observations strengthens the credibility of the findings. The careful selection and bias-correction of assimilated data ensure the quality and reliability of the simulations.

We would like to thank the reviewer for the positive feedback and the valuable comments and suggestions for improvement.

While the manuscript is robust, a few areas suggested below could benefit from further clarification:

P4, L105-107 mentions the use of weather diaries and records of eye observations regarding sunshine, cloudiness, precipitation, wind, and other variables. It would be helpful to (1) Specify how the qualitative descriptions are converted into comparable data points and any challenges faced during this process. (2) Clearly state how the digitized eye observations are integrated into the validation process of the simulations. Highlight any specific methodologies used to ensure the reliability of these qualitative data points.

This is a valuable point that we did not mention in the manuscript. We will include some explanations in Sect. 2.1. (P5, L142ff) as follows:

"Furthermore, we use eye observations from selected weather diaries in Switzerland that were recorded in semi-standardized terminology (Auchmann et al., 2012; St. Gallen / Meyer from CHIMES; Hari, 2021). Adjustments included manual re-coding of the available information into classes (e.g. bright, partly cloudy, overcast for cloud cover) or categories (e.g. spray, rain, snow for precipitation types). Refer to Auchmann et al. (2012), Brugnara et al. (2015) or Brönnimann (2023) for information on how to best re-code symbolic or word information, re-classifiy it, and attribute a plausible time to a record, among others.

Given the inherent uncertainties, we consider the final data points as being of qualitative, complementary and relative information which, when taken as a whole, may support or contradict our model outputs."

P5, Line 139-141, The manuscript mentions that the observation series are not homogenized, which could impact the reliability of the assimilated data. A discussion on the potential effects of this and any mitigation strategies would be beneficial.

Thank you for the suggestion. We will include a short explanation at P5 L141.

"Although we assume the raw records to be consistent over a period of only ten days, potential errors must be considered and interpreted when showing the raw data. For assimilation however, a simple correction approach was applied to correct for biases in the measurement series (see Sect. 2.4). In addition, the assimilation algorithms reject values that are too far off from the first-guess simulated value. Hence, negative effects of potentially erroneous values on the assimilation can be considered to be rather small to negligible."

P8, Line 173-174, the manuscript mentions the use of the 20CR as initial and boundary conditions for the downscaling experiments, but it does not specify the exact variables utilized. The distinction between variables used for downscaling and those used for data assimilation is crucial, and the manuscript does emphasize the use of data assimilation for pressure and temperature. However, it lacks clarity on which variables are used exclusively for downscaling. To improve clarity and completeness, the manuscript should explicitly list the variables read from the 20CR for dynamical downscaling. This list should differentiate between variables used for initial and boundary conditions in downscaling and those used in data assimilation.

It is indeed important to provide information on the variables from 20CR which serve as initial and boundary conditions. We will clarify this in Sect. 2.2 of the revised manuscript:

"Initial and boundary conditions of the regional simulations are taken from the 20CR ensemble mean. In particular, these encompass three–dimensional fields of temperature, humidity, geopotential height, pressure, and the horizontal components of wind speed, as well as two-dimensional fields of 2 m temperature and humidity, 10 m wind components, surface and sea level pressure, snow depth, skin temperature, sea surface temperature, a land/sea mask and a sea ice flag. Furthermore, four layers of soil temperature and soil moisture from 20CR are used to initialize the regional model."

P8, Line 182-184, the manuscript states that the WRF model employs three nested, limited-area domains with cell sizes of 27 km, 9 km, and 3 km. These domains are nested to refine the global information to regional and local scales. It could be helpful to describe the process of providing lateral boundary conditions for each nested domain. Specify how the data from the parent domains are used to initialize and drive the simulations of the nested domains. Consider including a diagram that illustrates the nesting process and the flow of lateral boundary conditions from the outermost to the innermost domains. This visual aid would help readers better understand the methodology.

Given the target audience of the journal, we agree on including a few more sentences on the actual flow of information from coarse to fine scales.

"In simple terms, the process of providing initial and lateral boundary conditions for each nested domain is as follows: For model initialization, the nested, smaller domain receives the information from the coarser domain at the horizontal and vertical coordinates that both domains share. For all other coordinates (and for the outermost domain which typically does not share exact coordinates with the reanalysis), this information is spatially interpolated to the finer grid cells. The simulations

are then incremented going forward in time until new information from the coarser domain is available, which is fed in at the lateral boundaries of the nested domain."

P8, Line 178, mentions "Here, we mainly use the mean of the 80 ensemble members." It is essential to introduce here how these ensembles are used, specifically whether they used the ensemble mean or individual ensemble members. In the later part of the manuscript, they indicate that the ensemble mean of the 80 ensemble members is primarily used for synoptic analyses and as initial and boundary conditions for the downscaling simulations. Considering the high variability on synoptic scale, can 80 ensemble mean on 6-hour time scale reflect the weather pattern specifically for June in 1816? It would be beneficial to discuss the implications of using the ensemble mean versus individual ensemble members. This could involve addressing the potential smoothing effects and how this choice impacts the simulation results and their interpretation.

Thank you for this suggestion. Whereas the full ensemble of 20CR is only used for veryfing the representation of the summer 1816 in this dataset (see Fig. S5 in the supplement), the other analyses presented in the manuscript are based on the 3-hourly ensemble mean. We will clarify this in the revised manuscript.

We also agree that it is important to explain the effects from using the ensemble mean in this study. In fact, we address this in the results section 3.1 on P12 L277ff, and we give an interpretation of using the ensemble mean versus members on P21 L496ff. In addition, we will insert a sentence about our previous experiences with smoothing effects, as well as related findings in published literature.

"Previous studies for the same region found some deviations in variables such as maximum wind speed from using the ensemble mean versus members, but small smoothing effects in the pressure fields and overall limited benefit from applying an ensemble approach when comparing to station observations (Stucki et al, 2015, 2016). Whereas regarding extreme events, smoothing effects may be more pronounced (e.g. Mahoney et al., 2022), the ensemble mean was found to provide accurate initial and boundary conditions, even if possibly less accurate than individual ensemble memebers (Michaelis and Lackmann, 2013)"

P13, Line 310, while Figure 2 effectively presents data for domains D02 and D03, it skips the outermost domain (D01) and the comparison with the 20CR data. Including D01 in the figure would help demonstrate the first step of the downscaling process, showing how well the WRF model captures large-scale atmospheric patterns compared to the 20CR data. This is crucial for validating the initial downscaling step and ensuring that the model accurately represents the broader atmospheric conditions before refining them in the nested domains. This also helps readers better understand the progressive refinement of the atmospheric data from the global scale (20CR) to the regional (D01) and local scales (D02 and D03), and demonstrates each step of the downscaling process enhances the transparency and credibility of the methodology. By the way, the 500 gph contours can not be observed in Fig2 (b) and (c).

Thank you for this suggestion. Referring also to the first reviewer's comments, we will indicate the borders of the three nested domains in figure 2a and add a panel to the figure showing D01 in the revised version. We will furthermore add the 500 gph contours in all panels.

P17, Line 392, Figure 5 presents meteograms comparing station observations and measurements with WRF model outputs for various variables over a period of time. While it provides comprehensive data, the figure appears too busy, making it difficult to easily interpret the comparisons. I suggest to simplify the figure to make it easier for readers to extract key information. Can group related variables together to provide a clear comparison and emphasize the most important differences between observations and model outputs.

A similar suggestion was made by Referee 1, so we realize that it would be better to reduce the information. For this, we will just keep the Bern location with the full information in one panel, and we will have a second panel with just the most illustrative values, and without the NODA information on it, see the planned figure caption below. Because we think of the mimicked meteograms as a comprehensive illustration, we will place the current figure with the full information in the Appendix. The text will be adapted to follow the new figure.

"Figure 5: Meteograms of station observations and measurements (red colors) for a) Bern (station id be01) with WRF NODA (ligher colors, grey, blue, orange) and DA (darker colors) simulations output for the nearest grid point, for the period between 4 and 12 June 1816 (x-axis).
The top panel in shows observed cloud cover (red squares from 'bright' with no fill to 'mixed' with cross and 'covered' with fill) vs. simulated low, mid- and upper level cloud fraction (larger bars indicate more cloudiness). The second row panel in a) shows simulations of downward short wave flux at ground surface (orange lines; W/m2) and relative humidity at 700 hPa (dashed blue line; permil).
The third row panel shows thrice-daily observations of precipitation vs. simulated precipitation (blue vertical bars, mm). The bottom panel shows observations of wind direction (red vectors at unit length; north is up) vs. simulated wind direction (grey wind vectors; north is up) and velocity (black line and vector length; m/s at 10 m above ground).
Panels b) are as in a), but for selected values for Zurich (zh00), Geneva (ge00) and St. Gall (sg01). Red vertical crosses (dark for observer Escher, light for Feer) indicate the relative humidity measurements on the ground for Zurich. Red vertical bars show precipitation measurements in mm for Geneva; darker red dots for Zurich stand for observer Escher, lighter for Feer. The red tilted cross means 'rain and snow' for St. Gall. Grey plus signs mean confirmed obserations, minus sign mean no record of precipitation found."

P23, in the summary and conclusions, the manuscript briefly touches on the successful application of the methodology for the Year Without Summer but does not delve deeply into the challenges and considerations for applying these methods further back in time. To enhance the manuscript's discussion on the wider application of their methodology in past climates, it would be beneficial to include a more detailed exploration of the challenges and potential strategies for overcoming them. This can be done in the discussion or conclusion sections.

We think that we have already mentioned the challenges and potiential in the conclusions section. However, we can make a clearer statement by adding one or two sentences, e.g.

"Given that global or regional gridded datasets may soon appear for periods beyond 1816 and more early measurements will become available, the prospects of soon entering the 18th century with such 4-dimensional studies are very good."

**References:**

Auchmann, R., Brönnimann, S., Breda, L., Bühler, M., Spadin, R., and Stickler, A.: Extreme climate, not extreme weather: The summer of 1816 in Geneva, Switzerland, Clim. Past, 8(1), 325–335, https://doi.org/10.5194/cp-8-325-2012, 2012.

Brönnimann, Stefan (2023). The weather diary of Georg Christoph Eimmart for Nuremberg, 1695–1704, Clim. Past, 19(7), 1345-1357, https://doi.org/10.5194/cp-19-1345-2023, 2023.

Brugnara, Y., Auchmann, R., Brönnimann, S., Allan, R. J., Auer, I., Barriendos, M., Bergström, H., Bhend, J., Brázdil, R., Compo, G. P., Cornes, R. C., Dominguez-Castro, F., van Engelen, A. F. V., Filipiak, J., Holopainen, J., Jourdain, S., Kunz, M., Luterbacher, J., Maugeri, M., Mercalli, L., Moberg, A., Mock, C. J., Pichard, G., Řezníčková, L., van der Schrier, G., Slonosky, V., Ustrnul, Z., Valente, M. A., Wypych, A., and Yin, X.: A collection of sub-daily pressure and temperature observations for the early instrumental period with a focus on the "year without a summer" 1816, Clim. Past, 11, 1027–1047, https://doi.org/10.5194/cp-11-1027-2015, 2015.

Hari, C.: An Evaluation of Meteorological Observations by Samuel Studer ( 1807-1818 ), MSc thesis, University of Bern, Switzerland, 127 pp., https://occrdata.unibe.ch/students/theses/msc/343.pdf, 2021.

Mahoney, K., McColl, C., Hultstrand, D. M., Kappel, W. D., McCormick, B., and Compo, G. P.: Blasts from the Past: Reimagining Historical Storms with Model Simulations to Modernize Dam Safety and Flood Risk Assessment, Bulletin of the American Meteorological Society, 103(2), E266-E280, https://doi.org/10.1175/BAMS-D-21-0133.1, 2022.

Michaelis, A. C. and Lackmann, G. M.: Numerical modeling of a historic storm: Simulating the Blizzard of 1888, Geophys. Res. Lett., 40(15), 4092–4097, https://doi.org/10.1002/grl.50750, 2013.

Stucki, P., Brönnimann, S., Martius, O., Welker, C. S., Rickli, R., Dierer, S., Bresch, D. N., Compo, G. P., Sardeshmukh, P. D., Romppainen-Martius, O., Welker, C. S., Rickli, R., Dierer, S., Bresch, D. N., Compo, G. P., Sardeshmukh, P. D., and Sardeshmukh, A. P. D.: Dynamical downscaling and loss modeling for the reconstruction of historical weather extremes and their impacts – A severe foehn storm in 1925, Bull. Am. Meteorol. Soc., 96(8), 1233–1241, https://doi.org/10.1175/BAMS-D-14-00041.1, 2015.

Stucki, P., Dierer, S., Welker, C. S., Gómez-Navarro, J. J., Raible, C., Martius, O., and Brönnimann, S.: Evaluation of downscaled wind speeds and parameterised gusts for recent and historical windstorms in Switzerland, Tellus. Series A - dynamic meteorology and oceanography, 68(1), 31820, https://doi.org/10.3402/tellusa.v68 .31820, 2016.